# Mechanical and Chemical Resistance of UV Coating Systems Prepared under Industrial Conditions

**DOI:** 10.3390/ma16124468

**Published:** 2023-06-19

**Authors:** Milena Henke, Barbara Lis, Tomasz Krystofiak

**Affiliations:** Department of Wood Science and Thermal Techniques, Faculty of Forestry and Wood Technology, Poznan University of Life Sciences, 60-627 Poznań, Poland; milena.henke@up.poznan.pl (M.H.); barbara.lis@up.poznan.pl (B.L.)

**Keywords:** UV lacquer system, Hg–Ga radiator, furniture, high-density fiberboard, resistance, scratching, impact, abrasion, cold liquid

## Abstract

In the upcoming years, it is expected that more furniture will be built from honeycomb panels due to the growing demand for lightweight, long-lasting furnishings. High-density fiberboard (HDF), previously used in the furniture industry as back walls in box furniture or drawer components, has become a popular facing material used in the production of honeycomb core panels. Varnishing the facing sheets of lightweight honeycomb core boards with the use of analog printing technology and UV lamps is a challenge for the industry. The aim of this study was to determine the effect of selected varnishing parameters on coating resistance by experimentally testing 48 coating variants. It was found that the interactions between the following variables were crucial in achieving adequate resistance: lamp power, the amounts of varnish applied, and the number of layers. The highest scratch, impact, and abrasion resistance values were observed for samples with optimal curing provided by more layers and maximum curing with 90 W/cm lamps. Based on the pareto chart, a model was generated that predicted the optimal settings for the highest scratch resistance. Resistance to cold liquids made with a colorimeter increases with lamp power.

## 1. Introduction

Forests are one of the most important renewable resources, providing many benefits to society and the economy. They play an important role in the ecosystem, constituting habitats for diverse wildlife, purifying the air, and preventing erosion. They help mitigate climate change by offsetting carbon dioxide emissions [1]. Approximately 30% of forests are used for the production of various wood-based materials [2]. However, economic transformations are contributing to the deforestation of the Earth. Over the past three decades, the rate of deforestation has decreased from 7.8 million hectares per year in 1990–2000 to 5.2 million hectares per year in 2000–2010 and 4.7 million in 2010–2020. The rate of net forest loss has slowed in the past decade [2]. Individual governments are introducing plans to increase forest area. The European Union saw an average increase of 2.6% in forest coverage between 1990 and 2020. Sweden was the only member state to observe a decline in forest coverage (−0.4%) [3]. Global roundwood production in the forestry sector continues to grow. From 2000 to 2019, annual harvesting increased from 3.5 billion cubic meters per year to 4 billion cubic meters, a 14% increase [4]. The most recent figures for European Union countries show a 2% decrease in roundwood harvesting in 2020 compared to 2019 [3].

Insufficient sources of wood and the very high cost of wood pose a challenge to the furniture industry [5]. It is worth noting that as much as 54% of the world’s forest resources are located within the borders of five countries: Russia (20%), Brazil (12%), Canada (9%), the USA (8%), and China (5%) [2]. World lumber production increased by 27% in 2019 compared to 2000 at 489 million cubic meters. There was a slight drop in 2019 following the 2018 record. The real boom is in the production of wood-based panels, which increased 109% over the same period and now stands at 373 million m^3^ [4]. In Europe and other parts of the world, the number and development of wood-based panel industry factories grew between 2011 and 2015. Recycling and products made from alternative raw materials are becoming more and more significant in the plate business. After being pre-shredded, the waste from thinning and the maintenance cuts left in the forest can be sent to wood-based panel manufacturing [5]. Material recovery trends are being supported by policies that encourage trash sorting and increased awareness. For both environmental and financial reasons, a large amount of these materials are being recycled back into the production process as secondary raw materials. These materials are an important component of particleboard. Particularly great opportunities lie in the use of waste paper, which is converted into cellular paper and used for the fillings of composite panels [6].

Particleboard, fiberboard, plywood, and veneers are all wood-based products that are less expensive than solid products [7]. Flexibility, biodegradability, compostability, and recyclability or incineration are some of the benefits that they bring [8]. Designers and manufacturers consider usability in addition to economic and environmental factors by selecting materials with clearly defined strength characteristics to lighten the weight of the furniture. Composites made of wood are the most widely used materials [9]. Wood-based composites are described as a combination of any wood-based materials glued together [10]. These include particleboard panels, MDF panels that are veneered with a veneer or HPL, or their lighter counterpart that is a paper core with hexagonal cells—called cellular paper—in a particleboard frame with facings of HDF, MDF, particleboard, or a veneer [11,12]. Composite technology using honeycomb recycled paper became widespread in the furniture industry in the 1990s due to its superiority in terms of the furniture’s lightness-to-strength ratio [13,14,15]. According to research conducted by firm Fact.MR, its CAGR indicator is expected to increase by 3.5% from 2020 to 2026; the CAGR for honeycomb boards will increase by 6.5% from 2021 to 2031. The market share of furniture with a honeycomb core is expected to increase due to the growing trend towards lightweight and durable materials. The low weight of honeycomb sandwich furniture reduces the price of transporting furniture [16]. In Asian countries, the technology has spread rapidly, and most furniture contains honeycomb paper-core board components. The Asia–Pacific region is expected to drive sales in the honeycomb furniture market [17].

The most important aspect from the customer’s point of view is the visual quality, so a high-quality coating is essential. The quality of coatings depends on many factors [9,18]. Varnish products for UV curing are based on acrylic copolymers and various types of hybrid solutions. They predominate in furniture manufacturing technologies [19,20]. The use of these products has reached a high level of technical solution, making it possible to produce coatings with a diverse range of esthetic and decorative features and functional properties [21,22,23]. Coating the substrate with varnish is the most popular method of improving the stability, durability, and appearance of materials. The surface quality of the board and the type of finishing material affect the overall quality of the product [24]. 

The suitability of varnish coatings under specific application conditions can be determined by testing the substrate’s adhesion and resistance to mechanical factors, primarily impact, scratching, and abrasion [25,26,27,28]. The hygroscopic properties of the boards are important when forming a bond between the substrate and the coating [24]. Many studies have been published on the resistance of coatings to mechanical factors and cold liquids on wood surfaces, such as beech (*Fagus sylvatica*), yellow birch (*Betula alleghaniensis*), black alder (*Alnus glutinosa*), oak (*Quercus robur*), turkey oak (*Quercus cerris*), ash (*Fraxinus excelsior*), walnut (*Juglans regia*), box elder (*Acer negundo*), black locust (*Robinia pseudoacacia*), horse chestnut (*Aesculus hippocastanum*), honey locust (*Gleditsia triacanthos*), and Chinese sumac (*Ailanthus altissima*) [24,29,30,31,32,33,34,35,36,37]. The surface resistance of wood-based panels has also been determined: MDF, particleboard, plywood [38,39], melamine-faced chipboard (MFC), and painted panel (Pp) [40]. There are few publications in the field of HDF surface finishing despite the great potential for developing HDF as a facing for honeycomb panels.

The goal of this study is to identify the varnishing variables that have the greatest impact on the resistance of the varnish coating on an HDF substrate. Two HDF boards of varying densities and from different factories were used for the study. The surface samples were prepared using different numbers of layers and applying different amounts of each varnish to the surface. In addition, the samples were varied by curing them with UV lamps of different powers. It was assumed that these factors would influence coating resistance the most. The companies producing furniture from lightweight composite panels with HDF facings are struggling to achieve adequate coating strength that will perform well in everyday use. So far, the effect of combined modifiable parameters during the varnishing process on coating strength has not been determined. Varnishing sandwich composite panels have great research potential, and the findings are desired by the furniture industry. 

## 2. Materials and Methods

### 2.1. Materials

The experimental material consisted of elements with dimensions of 700 × 390 × 22 mm made from board on frame with a honeycomb core (thickness: 18 mm) using a facesheet HDF (thickness: 2.5 mm) obtained from two different manufacturers (referred to as A and B; Table 1). Particleboard (thickness: 17 mm, density: 540 ± 20 kg/m^3^) was used for the frame’s construction (Kronospan, Szczecinek, Poland). 

With a 1.45 g/m^2^ honeycomb core as the core of the board, the compressive strength was 2.5 kPa at 23 °C and 50% humidity (Axxor, Zwolle, The Netherlands). The selected boards were glued together using a single-component PVAC adhesive (Synthos, Oświęcim, Poland) with a viscosity of 14,000 mPa·s, as measured with a Brookfield DV2T viscometer (Brookfield Engineering Laboratories Vertriebs GmbH, Lorch, Germany) at 40 ± 0.5 °C, and with a pH of 4.1, determined using a Metler Toledo S210 m (Metler Toledo, Greifensee, Switzerland). The honeycomb panels underwent the finishing process after a conditioning time of 96 h. Several layers of varnish systems were used, including a primer, basecoat, and topcoat varnish (Remmers Poland, Tarnowo Podgórne, Poland) (Table 2).

### 2.2. Surface Lacquer Finishing Process

The present study is a continuation of a gloss and color study conducted on the same variants [41]. For this study, 48 variants were selected. Different numbers of layers (five or six) and different amounts of varnish were applied. There were eight application variants (Table 3) applied on two types of HDF boards (Table 1). Three lamp powers (120, 90, and 60 W/cm) were used for curing. The honeycomb panels were prepared and finished under the technological conditions of the varnish line (Borne Furniture, Gorzów Wielkopolski, Poland).

Figure 1 shows a diagram of a finished sample, while Figure 2 shows the painting line on which it was produced.

In the first stage of the varnishing process, the elements were sanded. The Heesemann LSM8 + EA10 wide belt sander was used for the tests. Two abrasive belts with grain size P180-P220 and with corundum coating were used. Then, two layers of primer were applied in the amounts of 30 and 15 g/m^2^. These were cured with five mercury (Hg) lamps (Efsen UV & EB Technology, Holte, Denmark). A further operation was sanding (abrasive belt with grain size P360-P400) and the application of the basecoat varnish. The first layer of basecoat varnish was cured with two gallium-doped mercury (Ga) lamps (Efsen UV & EB Technology, Holte, Denmark); the next layer was applied in a fixed amount (30 g/m^2^) for all tested variants, after which the product was crosslinked with two Ga and Hg lamps. Some arrangements had three layers of basecoat paint cured with two Ga lamps. The last layer was the topcoat cured with three Ga lamps and five Hg lamps. All of these operations were performed on one production line at a speed of 50 m/min, an ambient temperature of 35 °C, and a relative humidity of 35%.

### 2.3. Determination of Scratch Resistance

A Clemen tester, described in PN-88, F-06100/11, was used to determine the scratch resistance. The instrument is composed of a moveable table (1), a rotary lever (2), weights (3), a handle with a phono needle (4), and a nut for balancing the surface (5) (Figure 3). Three samples measuring 100 mm × 100 mm were prepared for each variant. The samples were mounted on a table, and a knife with a blade protruding 0.8 mm was set on the surface. The table together with the sample was moved at a speed of 15 mm/s. Two measurements were taken on each sample [42].

### 2.4. Impact Test (Ball Method)

The impact resistance was evaluated using the presumptions outlined in ISO 4211-4. Figure 4 shows a cylindrical weight (B) dropped through a tube (A) onto a 14 mm diameter steel ball (C) from the specified height.

Five heights were selected for the test from which the weight was dropped. Five tests were performed for each height. The degree of damage to the test area was assessed by reference to a descriptive numerical rating code (Table 4). The largest diameter of the impact mark in each test area was measured with a Brinell magnifier [43].

### 2.5. Abrasion Resistance

The standard ISO 7784-3 was used to assess the abrasion resistance of the surface finishes. The test was performed on a Taber Abraser 5130 (Figure 5).

The abrasion test was performed on three samples for each variant. Before starting the test, the samples were properly prepared: the edges were secured with tape and then weighed. After the device was calibrated, the surface testing was conducted. CS-10 abrasive wheels were used. Each sample was weighed after 50, 100, 150, and 200 cycles. The weight loss was determined based on the difference between the weight before and after each test cycle [44].

### 2.6. Resistance to Cold Liquids

Surface resistance to cold liquids was determined according to the standard STN EN 12720 + A1. The test was performed on two randomly selected boards for each surface finish and wood substrate. Cold liquids, which are typical of everyday situations in households, were selected for the test (Table 5).

The test was carried out on conditioned samples at 23 ± 2 °C. Filter paper discs were immersed in the test liquids for 30 s before application with tweezers. The test surface was immediately protected against evaporation with an inverted petri dish. The test was performed for three time intervals: 10 min, 16 h, and 24 h. After the appropriate time interval, the petri dish and the disc were removed with tweezers. Any residual liquid was removed with absorbent paper. After 24 h, the surface was wiped with a cloth soaked in a cleaning solution and then washed with water. After 30 min, the test surface was rated according to Table 6 [45].

For selected samples in which the test trace was observed, the color was measured. Twin samples were checked for an applied color test. The color was evaluated according to the CIELAB system (EN ISO 11664-4, 2011) using a comparison colorimeter (DT-145) with an illumination/geometry angle of 45°/0°. The measuring aperture was 8 mm, and the light source (Illuminant) was D65. For each sample, 10 measurements were taken, and the average values were recorded. The color differences were determined with the delta E parameter, which indicates the total color difference [31,46,47].

### 2.7. Data Processing

Microsoft Excel and Minitab 19 software programs were used for the statistical analysis of the test results. In order to determine the influence of individual factors on the resistance of the surface, the normality of the distribution was checked, and the analysis of variance (ANOVA) was performed. The main effects chart was used to present the data. For scratch resistance, an effect Pareto chart was drawn to show the effect sizes. A prediction and optimization report was generated.

## 3. Results and Discussion

### 3.1. Resistance to Scratching

The normality of the distribution was verified using the Ryan–Joiner test, similar to the popular Shapiro–Wilk test. At the significance level of α = 5%, the collected scratch resistance measurement data were normally distributed for raw data, and the null hypothesis that the variance of the dependent variable error was equal in all groups was accepted at the set confidence level (>0.05) [48].

ANOVA was performed, and the data were evaluated based on five variables: the density of the HDF, the amount of basecoat applied, the amount of topcoat applied, the number of coats, and the different powers of the lamps used to cure the surface of the coating (Table 7). It was found that the average resistance values differed at the significance level of α = 0.05 and were statistically significantly different for the amount of basecoat, amount of topcoat, and lamp power (*p*-value < 0.05). The type of board and the number of applications were not statistically significantly different (*p*-value > 0.05).

A main effect screener for average outcomes was produced for the collected data. Based on the Pareto plot of the effect size in standard deviations (Figure 6), it was determined that the most influential factor was the amount of topcoat applied (larger than two). The version with a topcoat layer of 10 g/m^2^ (1695 N) resulted in the best effects; the application of 3 g/m^2^ was associated with 10% less resistance. The most effective variation had a lamp output of 90 W/cm (1625 N). When 120 W/cm lights were used, the results were 8% worse, showing that overcuring the coating has a negative impact on scratch resistance. Curing with lamps that had the lowest tested power (60 W/cm) resulted in a 3.5% decrease. Regarding the basecoat application, the variation with a layer of 45 g/m^2^ yielded the best resistance (1599 N). The variation that used a 40 g/m^2^ applied basecoat had a 6.5% worse outcome. With an increase to 50 g/m^2^, there was a decrease of 0.9% in scratch resistance. The type of board and the quantity of overlays used during the varnishing process were two factors that marginally affected the test’s outcome.

The crucial process variables were determined using a regression model (Figure 7). The results led to the identification of the variables that enabled the greatest scratch resistance. The model indicates that 40 g/m^2^ of primer and 10 g/m^2^ of topcoat should be applied and cured with a UV lamp set to 92 W/cm; this combination of parameters should produce a scratch resistance of a projected value of 1912 N. Based on the following model, the predicted scratch resistance can be determined.

Previously, various varnish coatings on MDF, particleboard, and plywood were tested. Based on these tests, the author came to the conclusion that a thicker coating increases resistance. This is consistent with the information provided on the improvement in quality when more topcoat is applied [38]. The application of a second layer of topcoat increases a coating’s resistance according to earlier experiments on wood samples with coatings made of numerous layers of waterborne varnishes [31].

However, the same principle was not used with the priming varnish. The functions of various varnishes provide the explanation. The chemical composition of a basecoat varnish is designed to provide the surface with color, evenness, and initial resistance. The fundamental goal of a topcoat is to guarantee a quality finish. According to reports, the ratios of the materials, the types of resin, and the properties of other additives all affect scratch resistance [30,38,39]. An effect of the bonding reaction formed between the coatings and the substrate has been noted, especially in the case of powder lacquers [38].

### 3.2. Impact Resistance

The normality of the distribution was verified using the Ryan–Joiner test, similar to the popular Shapiro–Wilk test. At the significance level of α = 5%, the data collected from the impact resistance measurements were normally distributed for raw data. ANOVA was performed, and the data were assessed on the basis of five variables: the density of the HDF, the amount of basecoat, the amount of topcoat, the number of layers, and the different lamp powers used to harden the surface of the coating. None of the results confirm a *p*-value > 0.05. This is characteristic of studies whose assessment is based on a multi-level scale. It was decided to present the evaluations in graphical form (Figure 8, Figure 9 and Figure 10).

For the tested heights (15, 25, 50, 100, and 200 mm) of the ball drop test, surface damage was observed and rated from 5 to 2 (Figure 8). As the height of the ball drop increased, the average rating decreased. However, this decrease was not proportional. Drops from heights of 25 and 50 mm caused similar damage, being 17% and 18% of the maximum rating, respectively. Another test performed at a height of 100 mm recorded 34% of the maximum and an average rating of close to 3. Doubling this height resulted in a 54% decrease in the rating to a level close to 2. The averages of the largest trace diameters were inversely proportional to the rating. The average trace diameters for 15 and 25 mm were similar with an 8% increase in diameter for a fall from a height of 50 mm. Subsequent heights of 100 mm and 200 mm resulted in 37% and 46% increases in diameter, respectively.

The study revealed that, for samples cured with UV lamps at 60 and 90 W/cm, the resistance rating dropped as the amount of topcoat increased (Figure 9). Comparing variants with the lamp power set to 60 W/cm between topcoats of 3 and 10 g/m^2^, the average resistance rating in the 200 mm ball impact test decreased by 20.83%. For a lamp power of 90 W/cm, the resistance rating (200 mm) decreased by 25%. For samples cured with 120 W/cm UV lamps, the relationship was the opposite: the rating increased by 16.67% (200 mm) with a thicker topcoat. The results indicate the importance of correctly adjusting the lamp power in relation to the amount of topcoat applied. Increasing the amount of topcoat without changing the UV lamp power or increasing the lamp power without changing the application will result in less impact resistance because the surface will be insufficiently cured or processed. Increasing the application and lamp power will improve impact resistance.

A higher resistance rating is also affected by the number of layers used in the varnishing process (Figure 10). More layers of varnish result in better curing of each layer. The amount of primer varnish and the type of substrate did not affect the results.

The results correspond with those of tests conducted on other types of varnish. An effect of curing cross-linking density on coating resistance was found. Coatings with the highest cross-linking density achieved the highest results [24].

Researchers determined the resistance of waterborne varnishes. They observed poor impact resistance for the products [49]. Coatings cracked even at a drop of 10 mm on a wooden substrate. One factor that had a positive effect on coating resistance was the use of two-component solutions [32]. In the case of a 200 mm drop, the resistance of the coating on an MDF substrate was rated at level 2, similar to the HDF board in this study.

The influence of the interaction between the coating and the wood substrate was investigated [31]. It was found that, with the help of varnish additives, the mechanical properties of the coating can be improved, including impact resistance [30].

### 3.3. Abrasion Resistance

The normality of the distribution was verified using the Ryan–Joiner test, similar to the popular Shapiro-Wilk test. At the significance level of α = 5%, the data collected from abrasion resistance measurements were normally distributed for raw data. A main effects plot was created based on the results gathered after 50, 100, 150, and 200 sample revolutions (Figure 11). Weight loss increased with the passing of subsequent cycles. For the next tests, the additional weight loss was equal, amounting to 1.2–1.4 mg.

One of the elements that distinguished the coatings’ quality was the number of applications. A smaller weight loss was noted from more applications for each of the tested cycles. The middle layer of the coating, or the primer, is applied in three layers rather than two, although the total amount of varnish used remains the same. Applying and curing smaller amounts while increasing the number of layers had a positive effect.

The amount of basecoat is the second component, and it is closely related to the previous one. The weight loss decreased as the basecoat thickness increased. Initially, for 50 and 100 cycles, the difference in weight loss between the applications of 40 and 45 g/m^2^ was greater than the difference between 45 and 50 g/m^2^. It was only after the 150-cycle test that this difference decreased. On the other hand, the use of 200 cycles showed a different trend. The coating with 50 g/m^2^ lost weight at a rate that was 1.9 mg lower than the coatings with 45 g/m^2^ of varnish applied.

For the topcoats, applying 10 g/m^2^ decreased the weight loss from the 50-, 100-, and 150-cycle tests. The trend reversed for 200 cycles. A greater amount of weight was lost with the application of 10 g/m^2^ than with 3 g/m^2^.

It is clearly demonstrated that the larger density provided a smaller loss due to abrasion by analyzing the impact of the type of board. Fifty cycles of abrasion led to similar results for different UV lamp powers. The smallest weight loss was observed in abrasion testing with 100, 150, and 200 cycles when using the middle type of curing with 90 W/cm lamps. The greatest amount of weight was lost when 120 W/cm UV lamps were used, proving the negative influence of overly strong hardening. A suitable amount of varnish products—50 g/m^2^ of basecoat and 10 g/m^2^ of topcoat—applied in numerous, thinner layers proved to make an abrasion-resistant board.

These findings are consistent with tests of waterborne and powder varnish coatings performed on MDF, particleboard, and plywood. According to previous research, a thicker coating leads to improved abrasion resistance [38]. A second topcoat layer proved to increase the number of revolutions needed to reach the initial point of wear (IP) by 28% in testing of waterborne lacquer coatings on a wood substrate [32]. The degree of coating hardening has a significant impact on a coating’s resistance according to researchers [50]. MDF has the lowest resistance to abrasion among the well-known and tested wood-based boards. The high near-surface density of MDF boards required only a small amount of varnish to be applied. The porosity of the material was linked to particleboard’s greater resistance [38]. Researchers looked at several types of wood and reached similar conclusions [30]. An earlier study focused on HDF boards’ roughness. After sanding, there was only a 0.25 µm variation in the Ra parameter as measured with a Mitutoyo SJ-210 profilometer [51]. Additionally, it is thought that technological factors in manufacturing, such as the type and concentration of glue, the pressure, and the temperature used to produce the boards, interact with the painted surfaces [38].

### 3.4. Resistance to Cold Liquid

Among the liquids tested, wine, coffee, and acetone left a mark (Table 8). No coating changes were observed in the test within 10 min. After 16 h, the color had changed on a few samples. For wine and coffee, each of these samples was cured with 60 W/cm of lamp power. Only two of the eight variations cured with 60 W/cm had no visible color change. The samples with no differences in color were varnished with more layers, and the total amount of primer varnish was 45 or 50 g/m^2^. In the acetone case, the amount of topcoat had the greatest influence on color change. Samples with more topcoats were more prone to color changes. The exception was the variant where the lamp power was 90 W/cm. In the samples with acetone, no difference was observed between the 16 and 24 h tests. In the case of wine and coffee, a significant increase in the number of color changes was observed. The amount of color changes on samples with wine was 41.5% higher than for the variant after 16 h. Color changes did not occur on the samples with 10 g/m^2^ of a topcoat. Two variants cured with 60 W/cm were an exception. In the case of coffee after 24 h, all samples had changed color, which was 75% higher than for the 16 h test.

The decision was made to evaluate color changes using a colorimeter for coffee and wine stains after 24 h because the previous method of visually assessing the chemical resistance was subjective, and all damaged variations were given the same score (4). The ΔE parameter describes the variation in color between the test sample and the control sample.

The outcomes demonstrate that the degree of curing for a coating is a crucial factor that determines its chemical resistance (Figure 12). The ΔE parameter decreased as lamp power increased during curing. In the case of wine, there was a difference of 23% between the two variants of 60 and 90 W/cm and 34% between 90 and 120 W/cm. The difference for coffee was substantially greater: 56% between variants cured with 60 and 90 W/cm and 45% between those with 90 and 120 W/cm.

The ΔE value decreased as the UV lamp power increased. Therefore, in the case of coffee, the degree of hardening of a coating can be considered a determining factor for the chemical resistance of the coating (Figure 13). Applying a 3 g/m^2^ topcoat and curing the coating with 120 W/cm lamp power resulted in the highest resistance. The variant in which the surface was cured with the same lamp power and the topcoat was applied at a thickness of 10 g/m^2^ obtained a 5% worse result. The variants cured with 60 and 90 W/cm of lamp power maintain the same relationship: higher resistance was obtained with less topcoat being applied. The difference between the versions when using UV lamps set at 90 W/cm was 43.3%, while at 60 W/cm, it was 26.6%. The smallest difference was observed when the coating was cured with a lamp power of 120 W/cm. Because the surface in both variants was sufficiently cured, the change in application did not significantly impair the chemical resistance. In summary, the thinner the varnish application and the higher the UV lamp power, the lower the value of ΔE, i.e., resistance to cold liquids.

The results obtained for the resistance of coatings did not indicate an effect from the type of substrate on the resistance to cold liquids in the range of HDF boards of different densities. To date, no effect of wood species has been observed on coating resistance. Only the type of finished coating determines its resistance to marks. Wine left traces on the coatings made of a single-component waterborne acrylic/polyurethane varnish, tung oil [31]. Condensed milk and disinfectant/sanitizer stained the surfaces with a single or double layer of linseed oil [36]. The surfaces coated with silicone varnishes had low resistance to most of the liquids tested with the exception of alcohol [37]. Sodium hydroxide left marks on the synthetic varnish coatings, and acetic acid left marks on the polyester and cellulose varnish. Acetone proved to be a critical point for the three varnishes mentioned above as well as the polyurethane varnish. Nannolacke UV varnish showed a higher liquid resistance than the polyurethane, polyester, cellulosic, and synthetic varnishes [35]. Alcohol-induced staining of the coating was observed on both MDF and wood [33,52], which directly corresponds to the results obtained. In the case of waterborne varnish testing, an effect from the topcoat on the coating’s resistance was observed [32]. The above studies performed with a colorimeter developed the conclusion that not only the type but also the amount of topcoat applied determine the discoloration.

## 4. Conclusions

Based on simulations and experiments, it was found that four out of the five studied parameters are crucial for the resistance of furniture varnish coating.

UV lamp power is the parameter with the greatest impact on chemical resistance. The research indicates that, in order to achieve the least significant discoloration, it is necessary to maximize the power of the UV lamps. The degree of surface curing is also a very important parameter in the case of scratch and impact resistance, but in these cases, it is necessary to adjust the power of the UV lamps to the amount of varnish applied in order to obtain the highest resistance. In this study, the highest resistance was obtained when the UV lamp power was set to 90 W/cm.

Another important parameter is the number of layers of basecoat applied. An additional layer of basecoat improves the furniture’s abrasion and impact resistance. The application of another layer of coating is associated with higher electricity costs in the production of furniture components, but it results in a significant improvement in the above-mentioned characteristics due to the fact that each layer of varnish is thinner and better cured.

Besides the number of layers of basecoat, the amount of basecoat applied is also important, especially for scratch and abrasion resistance. In the experiments, the worst results were observed from the variant with the least primer applied (40 g/m^2^), while the other variants were at a similar level. In the case of abrasion resistance, a thicker application resulted in better resistance in every variant tested.

The amount of topcoat applied is the most important factor in terms of scratch resistance. By applying a thicker topcoat layer, scratch resistance can be improved.

The density of the HDF boards had no significant effect on the mechanical resistance of the varnish coating in any of the tests conducted.

## Figures and Tables

**Figure 1 materials-16-04468-f001:**
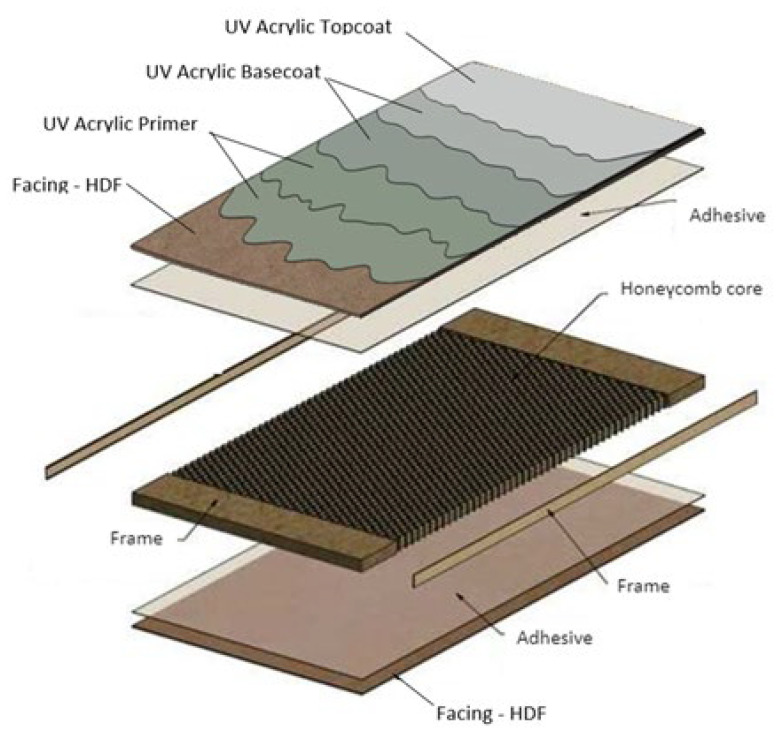
An overview of the material’s structure.

**Figure 2 materials-16-04468-f002:**
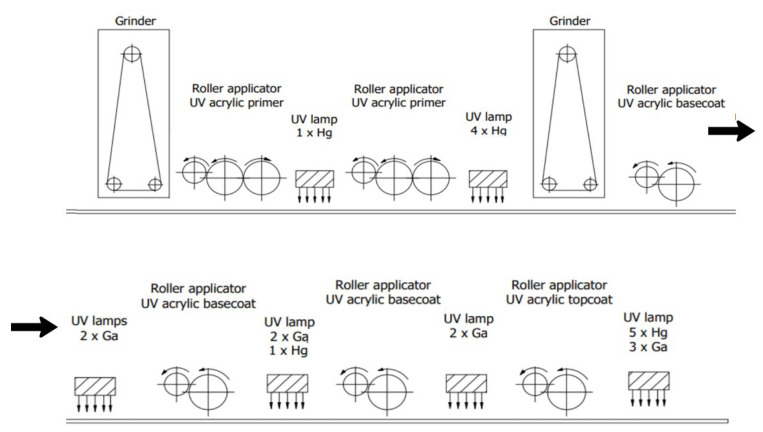
An overview of the varnish line on which the tests were conducted.

**Figure 3 materials-16-04468-f003:**
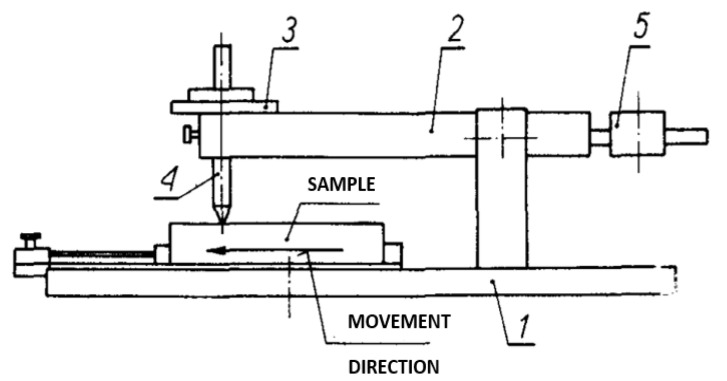
Clemen tester [42].

**Figure 4 materials-16-04468-f004:**
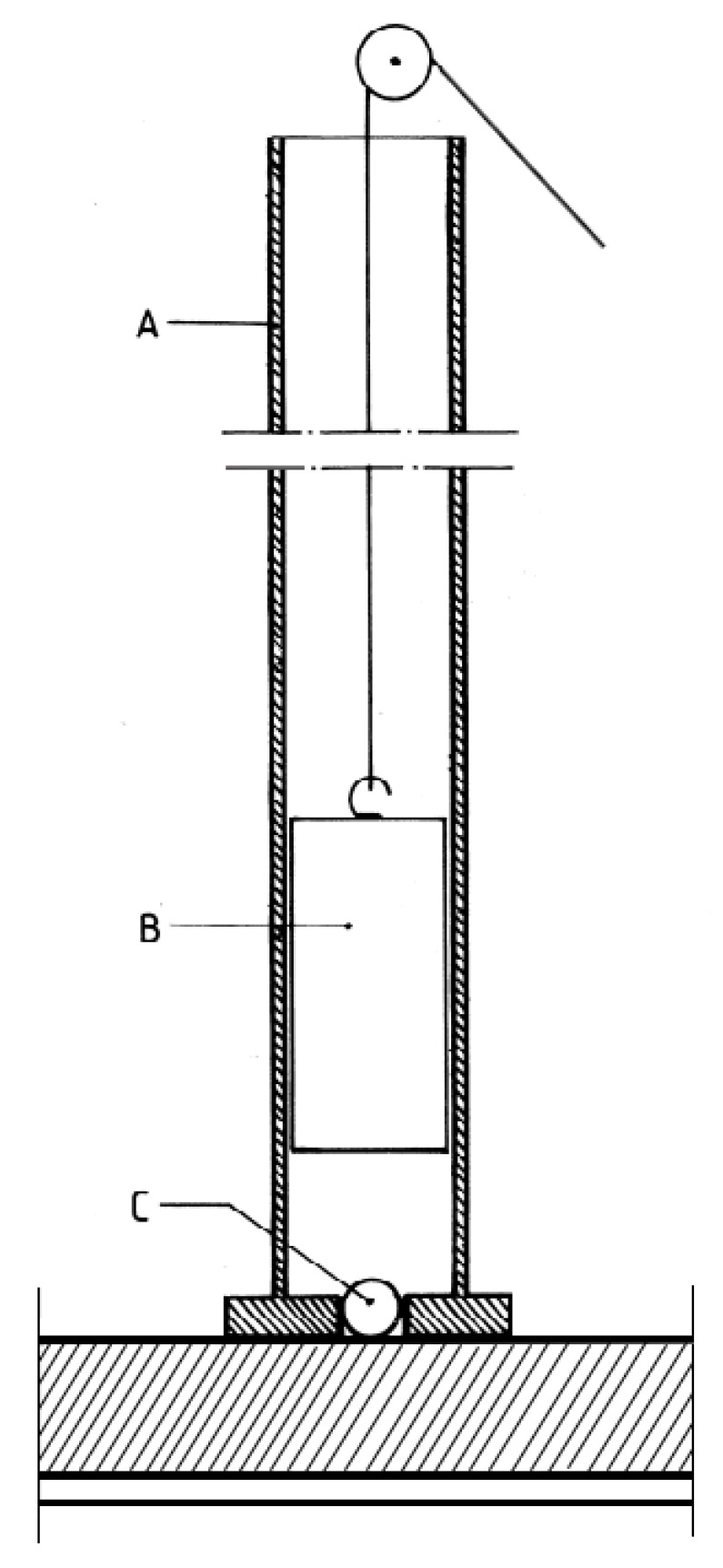
Ball impact method [43].

**Figure 5 materials-16-04468-f005:**
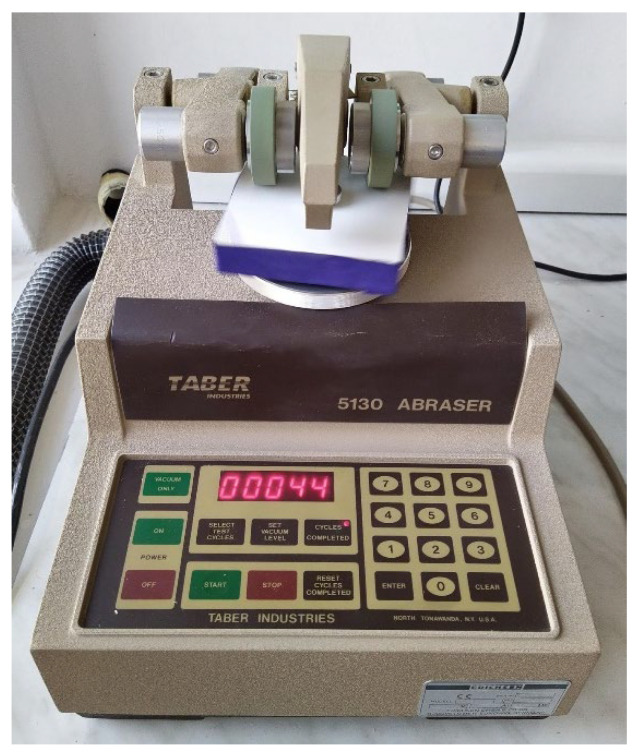
Taber Abraser test.

**Figure 6 materials-16-04468-f006:**
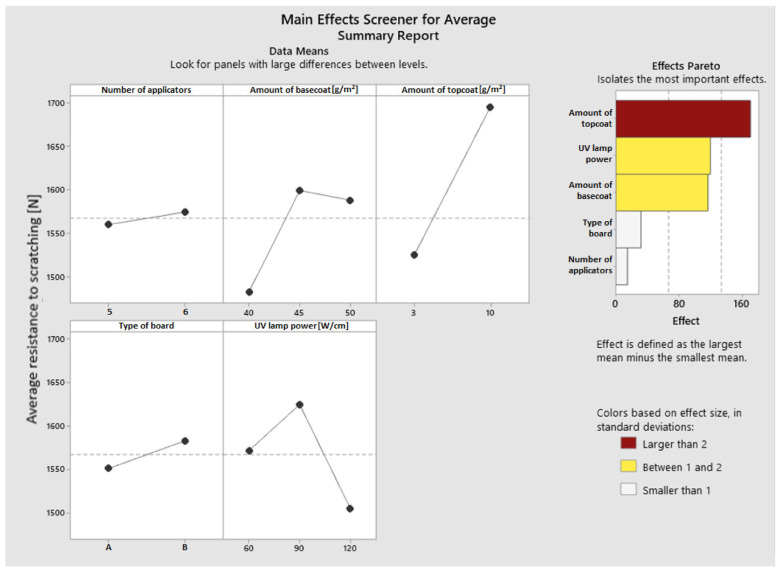
Main effects screener for average resistance to scratching.

**Figure 7 materials-16-04468-f007:**
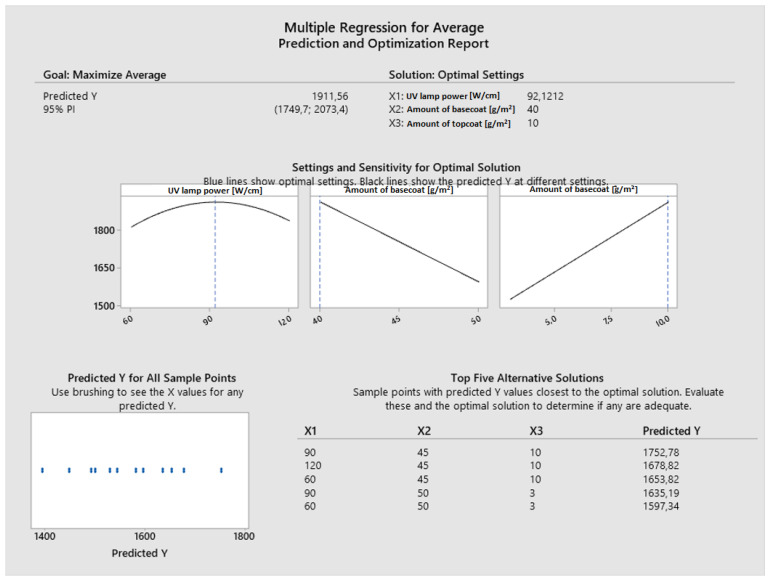
Multiple regression for average scratching.

**Figure 8 materials-16-04468-f008:**
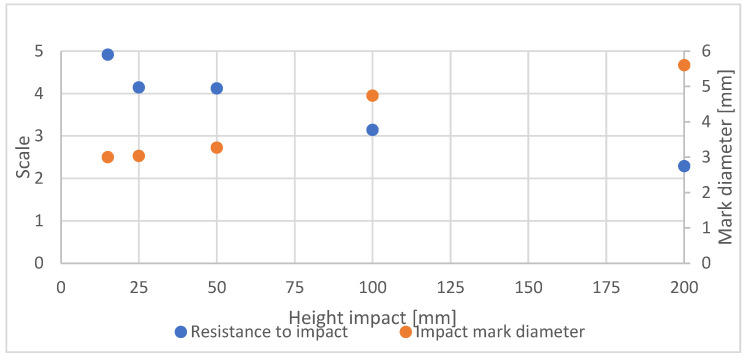
Average impact resistance rating by height.

**Figure 9 materials-16-04468-f009:**
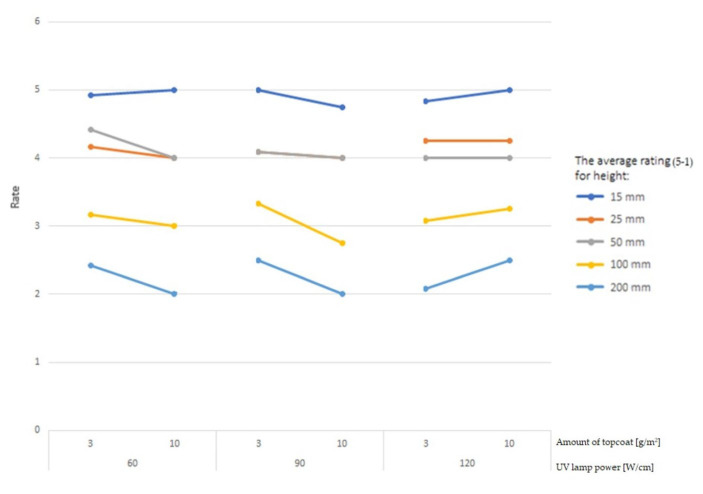
Average rating according to the amount of topcoat used and the power of the lamps.

**Figure 10 materials-16-04468-f010:**
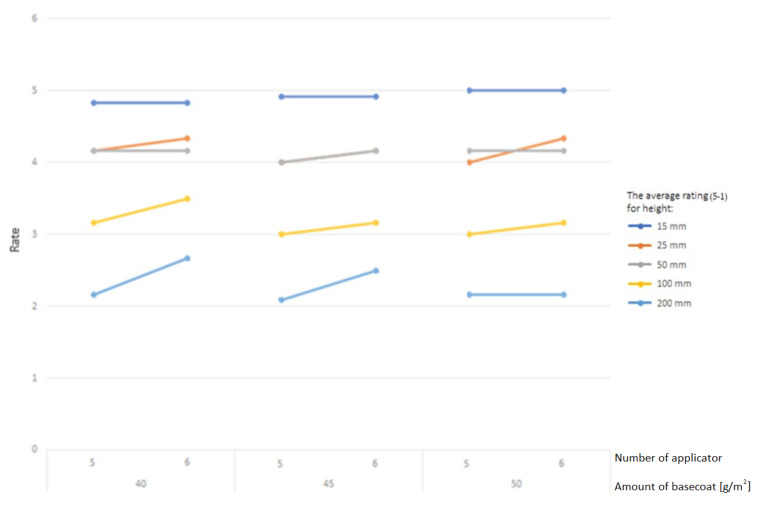
Average rating according to the amount of basecoat used and the number of applications.

**Figure 11 materials-16-04468-f011:**
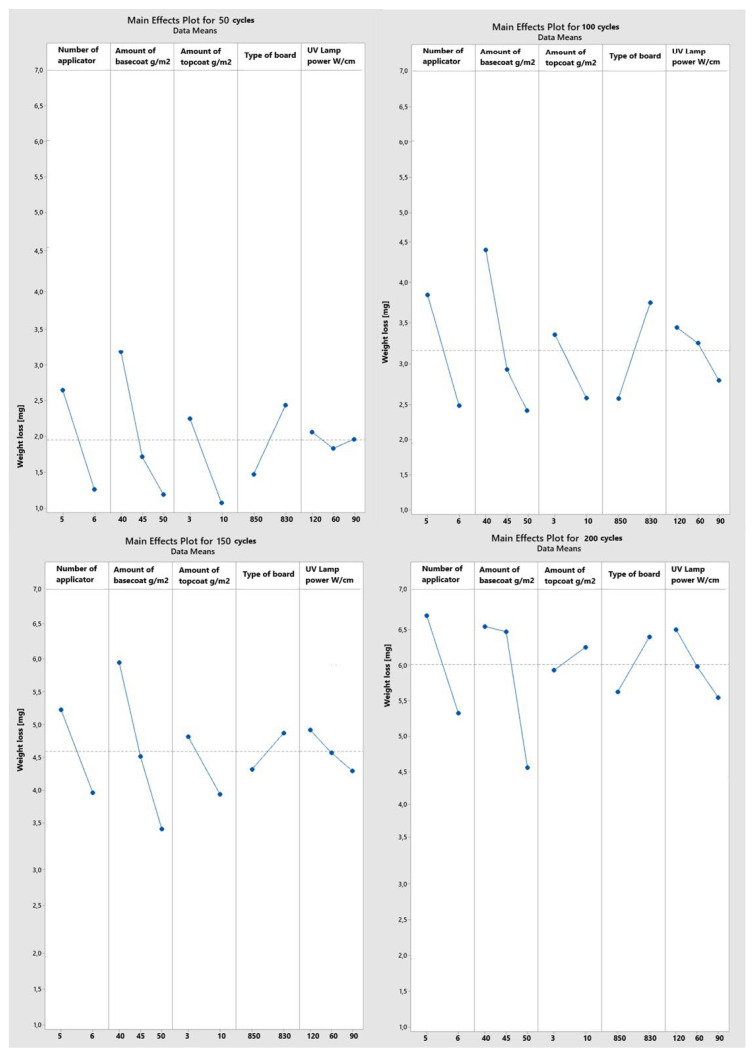
Main effects plots for weight loss after 50, 100, 150, and 200 cycles.

**Figure 12 materials-16-04468-f012:**
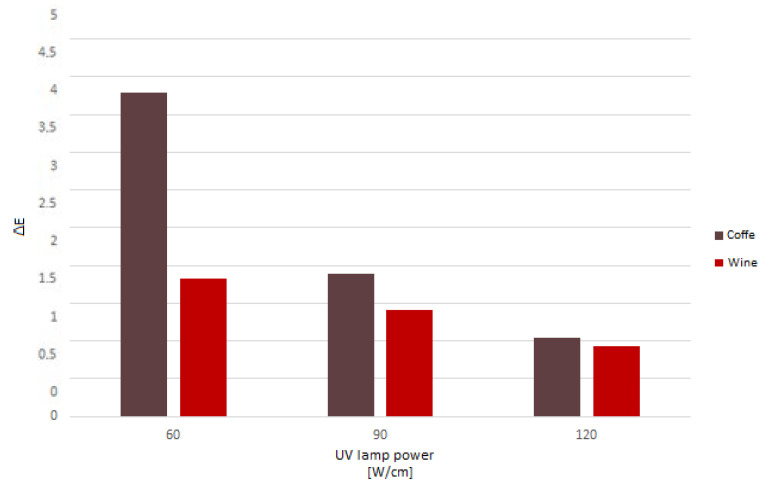
Results of the ΔE parameter for the tested coffee and wine marks after the 24 h test.

**Figure 13 materials-16-04468-f013:**
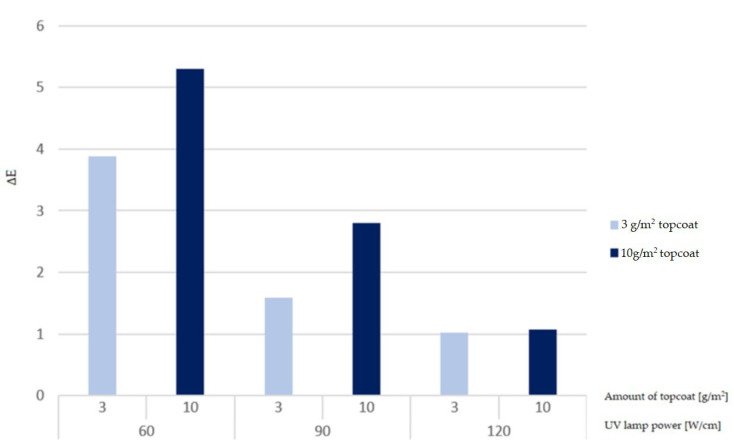
Results of the ΔE parameter for the tested coffee marks after the 24 h test.

**Table 1 materials-16-04468-t001:** Basic properties of the HDF boards.

Parameter	Board Label	
	A	B
Density [kg/m^3^] according to DIN EN 323:1993	850	830
Modulus of elasticity [MPa] according to DIN EN 310:1993	4300	4500
Humidity [%] according to DIN EN 322:1993	7	7
Swelling resistance [%] according to DIN EN 317:1999	45	45

**Table 2 materials-16-04468-t002:** Basic physicochemical properties of the varnish products.

Parameter	UV Primer	UV Basecoat	UV Topcoat
Density [g/cm^3^]	1.63 ± 0.15	1.73	1.30 ± 0.15
Solid content [%] according to PN-EN ISO 3251:2019	95.3 ± 0.5	98.3 ± 0.5	97.8 ± 0.5
Viscosity [mPa.s] (Brookfield, Thermosel 35 °C, 20 rpm, spindle 27)	7700	400	1475

**Table 3 materials-16-04468-t003:** Amounts of varnish applied in particular variants.

Variants of Application									
Type of Varnish Product	Number of Layers	Amount of Varnish Applied [g/m^2^]							
UV acrylic primer	1	30	30	30	30	30	30	30	30
	2	20	20	20	20	20	20	20	20
UV acrylic basecoat	1	15	7.5	7.5	10	5	10	15	20
	2	30	30	30	30	30	30	30	30
	3		7.5	7.5	10	5			
UV acrylic topcoat	1	10	10	3	3	3	3	3	3

**Table 4 materials-16-04468-t004:** Ball impact evaluation criteria.

Rating	Criteria
5	No visible marks on the surface
4	No cracks on the surface, but an impact mark is visible only when the light from a light source is reflected off the test surface at or quite close to the test point back to the observer’s eyes
3	Slightly cracked surface, generally one or two circular cracks around the impact mark
2	Moderate to heavy crack formation within the limits of the impact mark
1	Crack formation beyond the impact mark and/or flaking of the surface finish or surface covering material

**Table 5 materials-16-04468-t005:** Cold liquids used.

Cold Liquid	Characteristic
Distilled water	-
Acetone	-
Paraffin	Paraffinum liquidum
Ethylene	48% (*v*/*v*) aqueous solution
Wine	Merlot Trevenezie IGT 2021
Tea	1.75 g of tea leaves infused in 175 mL of boiling water, leached for 5 min without stirring, and then carefully decanted
Coffee	40 g of instant, freeze-dried coffee dissolved in 1 L ofboiling water
Beetroot juice	100% beetroot juice (Biurkom Flampol Sp. z o.o., Poland)
Blackcurrant juice	Pasteurized nectar, blackcurrant juice from concentrated juice (26%), fruit content minimum 26%, (Tymbark-MWS Sp. z o.o., Poland)
Condensed milk	8% fat content, sweetened (Milk Company in Gostyn, Poland)

**Table 6 materials-16-04468-t006:** Surface evaluation criteria.

Degree	Description
5	No visible changes (no damage)
4	Slight change in gloss—visible only in the reflection of a light source, e.g., discoloration or change in color or gloss; no change in the surface structure, e.g., swelling, fiber elevation, cracking, or blistering
3	Slight traces of damage (gloss)—visible from multiple perspectives, e.g., discoloration or change in color or gloss; no change in the surface structure, e.g., swelling, fiber elevation, cracking, or blistering
2	Strong traces of damage—visible in all viewing directions, e.g., discoloration, change in color or gloss, and/or the surface structure has changed slightly, e.g., swelling, fiber elevation, cracking, or blistering
1	Strong damage—the surface structure has changed noticeably and/or discoloration or change in color or gloss, and/or the surface material has partially or completely come off, and/or the filter paper sticks to the surface

**Table 7 materials-16-04468-t007:** One-way ANOVA of scratch resistance as a function of the process variables.

One-Way ANOVA Response	Source	DF	Adj SS	Adj MS	F-Value	*p*-Value
	UV Lamp power [W/cm]	2	115,289	57,645	4.89	0.012
	Amount of topcoat [g/m^2^]	1	261,235	261,235	31.24	0.000
Scratch resistance	Application of basecoat [g/m^2^]	2	115,833	57,917	4.92	0.012
	Number of applications	1	2552	2552	0.18	0.671
	Type of board	1	12,245	12,245	0.89	0.351

**Table 8 materials-16-04468-t008:** Evaluation of the cold liquid resistance test after 16 and 24 h, according to Table 6. X—Degree 4: Slight change in gloss, visible only in reflection of light source; •—Degree 5: No visible changes.

Type of Varnish Product	Number of Layers	UV Lamp Power 60 W/cm								UV Lamp Power 90 W/cm								UV Lamp Power 120 W/cm							
		Amount of Varnish Applied, in g/m^2^								Amount of Varnish Applied, in g/m^2^								Amount of Varnish Applied, in g/m^2^							
UV acrylic primer	1	30								30								30							
	2	20								20								20							
UV acrylic basecoat	1	15	7.5	7.5	10	5	10	15	20	15	7.5	7.5	10	5	10	15	20	15	7.5	7.5	10	5	10	15	20
	2	30	30	30	30	30	30	30	30	30	30	30	30	30	30	30	30	30	30	30	30	30	30	30	30
	3		7.5	7.5	10	5					7.5	7.5	10	5					7.5	7.5	10	5			
	Sumaric	45	45	45	50	40	40	45	50	45	45	45	50	40	40	45	50	45	45	45	50	40	40	45	50
UV acrylic topcoat	1	10	10	3	3	3	3	3	3	10	10	3	3	3	3	3	3	10	10	3	3	3	3	3	3
Evaluation of surface resistance to cold liquids	Acetone 16 h	X	X	•	•	•	•	•	•	•	•	•	•	•	•	•	•	X	X	•	•	•	•	•	•
	Wine 16 h	X	•	X	•	X	X	X	X	•	•	•	•	•	•	•	•	•	•	•	•	•	•	•	•
	Coffee 16 h	X	•	X	•	X	X	X	X	•	•	•	•	•	•	•	•	•	•	•	•	•	•	•	•
	Acetone 24 h	X	X	•	•	•	•	•	•	•	•	•	•	•	•	•	•	X	X	•	•	•	•	•	•
	Wine 24 h	X	X	X	X	X	X	X	X	•	•	X	X	X	X	X	X	•	•	X	X	X	X	X	X
	Coffee 24 h	X	X	X	X	X	X	X	X	X	X	X	X	X	X	X	X	X	X	X	X	X	X	X	X

## Data Availability

Department of Wood Science and Thermal Techniques, Faculty of Forestry and Wood Technology, Poznan University of Life Sciences, Wojska Polskiego St. 38/42, 60-627 Poznan, Poland; milena.henke@up.poznan.pl.

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
