# Peer review of "Mechanical and Chemical Resistance of UV Coating Systems Prepared under Industrial Conditions"

_materials, 2023, doi:10.3390/ma16124468_

Round 1

Reviewer 1 Report

The manuscript regards the detail study on Resistance upon mechanical and chemical factors of UV coating systems prepared under industrial conditions. It has been well prepared, however some revisions are required:

1. English can be improved

2. Many typos should be corrected. For example: Line number/L 35, L 159, L 226, etc., please check for the whole text in the manuscript

3. Please improve the quality of the figures (Figs. 4, 9, 10, 12, 13)

4. Introduction: please add information on the industrial conditions of UV coating systems that commonly used in the furniture industries

5. L 122 : density of 40 +/- 20 kg/m3 should be 400 +/- 20 kg/m3

6. L 272: "The fundamental goal of topcoat is to guarantee a quality finish": difficult to understand, please revise

7. L 330: Figure 81, please correct to Figure 11

8. Conclusion: Suggestion on the optimum parameter conditions of UV coating based on this research, to be applied for industry, might be added in the Conclusion

9. References: Ref.15 & 48 (L 506 & L 575): please add the publication sources

Please correct the typos and improved the grammar

Author Response

Dear Reviewer,

first of all thank you very much for your kind words and useful comments about our paper. Below our answer:

The manuscript regards the detail study on Resistance upon mechanical and chemical factors of UV coating systems prepared under industrial conditions. It has been well prepared, however some revisions are required:

Answer:

Thank you very much for your kind words.

  1. English can be improved

Answer:

We agree with the reviewer. The text has been professionally proofread.

  1. Many typos should be corrected. For example: Line number/L 35, L 159, L 226, etc., please check for the whole text in the manuscript

Answer:

We agree with the reviewer. Typos were corrected.

  1. Please improve the quality of the figures (Figs. 4, 9, 10, 12, 13).

Answer:

The figures have been prepared in the special program. There is a big problem to significantly improve the quality of the figures.

  1. Introduction:please add information on the industrial conditions of UV coating systems that commonly used in the furniture industries

Answer:

The research was carried out on a professional production line using innovative solutions. Many technological details are covered by a confidentiality clause. Therefore, only the necessary information is provided in the article.

  1. L 122 : density of 40 +/- 20 kg/m3should be 400+/- 20 kg/m3

Answer:

Changed in the text to 540 +/- 20 kg/m3

  1. L 272: "The fundamental goal of topcoat is to guarantee a quality finish": difficult to understand, please revise.

Answer:

The sentence is after professional proofreading. We wanted to give readers information that in the multilayer system the top coating is responsible for the final resistance of coatings.

  1. L 330: Figure 81, please correct to Figure 11

Answer:

Done

  1. Conclusion:Suggestion on the optimum parameter conditions of UV coating based on this research, to be applied for industry, might be added in the Conclusion

Answer:

This paper is a part of the PhD. After next investigations in the next papers we will select optimum parameter conditions of UV coating.  

  1. References: Ref.15 & 48 (L 506 & L 575): please add the publication sources –

Answer:

Were added

Reviewer 2 Report

I find the article very interesting and provides knowledge. Well structured and clarifying.

Furniture made from honeycomb panels is increasingly in demand. High-density fiberboard (HDF) is also increasingly in demand as a covering material. The article deals with a current issue. It also presents as a novelty the varnishing of these boards using UV lamps. The results help the industry by being able to apply this method, for example, obtaining good results in terms of scratches.

Only modify some formal aspects.

In table 5, correct coffe by coffee.

Delete the titles inside figures 8, 9, 10, 12, 13.

In figure 7 there is text overlay.

In figures 9 and 10 separate the value (mm) from the unit.

Determine the effect of the selected varnishing parameters on the resistance of the coating by experimentally testing 48 coating variants, considering that it is sufficient. Probably the authors can present studies of durability, temperature, humidity in later articles.

Author Response

Dear Reviewer,

first of all thank you very much for your kind words and useful comments about our paper. Below our answer:

Furniture made from honeycomb panels is increasingly in demand. High-density fiberboard (HDF) is also increasingly in demand as a covering material. The article deals with a current issue. It also presents as a novelty the varnishing of these boards using UV lamps. The results help the industry by being able to apply this method, for example, obtaining good results in terms of scratches.
Only modify some formal aspects.
Answer:

Thank you very much for your kind words.

In table 5, correct coffe by coffee.
Answer:

Was corrected

Delete the titles inside figures 8, 9, 10, 12, 13.
Answer:

Were deleted

In figure 7 there is text overlay.

Answer:

We used a special program for analysis. It is difficult to make changes to the original figure.

In figures 9 and 10 separate the value (mm) from the unit.
Answer:

Was corrected

Determine the effect of the selected varnishing parameters on the resistance of the coating by experimentally testing 48 coating variants, considering that it is sufficient.

Probably the authors can present studies of durability, temperature, humidity in later articles.

Answer:

We strongly agree with your comment. In the next paper (-s) we will present results of durability of coatings prepared in the industrial conditions.

Reviewer 3 Report

The reviewed paper is very interesting to the readers. Authors describe effect of selected parameters on coating resistance on HDF material. I have few comments.

- line 72: I think there is a mistake in "by Fact.MR"

- line 131: word "panles"

- the quality of fig. 4 should be improved and there should be described what is A, B, C

- Figure 6: I thing Y axis (Average) should have dimensiong (variable and units)

- The change in color can be described by defined criteria (invisible changes, small changes etc.). Is this applicable for investigated color change?

I think that article can be published after minor revisions.

Author Response

Dear Reviewer,

first of all thank you very much for your kind words and useful comments about our paper. Below our answer:

The reviewed paper is very interesting to the readers. Authors describe effect of selected parameters on coating resistance on HDF material. I have few comments.

Answer:

Thank you very much for your kind words.

- line 72: I think there is a mistake in "by Fact.MR"  

Answer:

Fact.MR is a fast-growing market research firm. Fact.MR offers actionable market insights, custom market analysis and consulting services across U.S, Europe and Asia.

- line 131: word "panles"

Answer:

Was corrected

- the quality of fig. 4 should be improved and there should be described what is A, B, C

Answer:

We strongly agree with the reviewer. We added information to the text: “Figure 4 shows a cylindrical weight (B) dropped through a tube (A) onto  a 14 mm diameter steel ball (C) from the specified height”.

- Figure 6: I thing Y axis (Average) should have dimensiong (variable and units)

Answer:

Done

- The change in color can be described by defined criteria (invisible changes, small changes etc.). Is this applicable for investigated color change?

Answer:

We used dots and crosses to describe the criteria. In the next article (-s) we will mainly analyse the colour changes after the accelerated aging test.

Reviewer 4 Report

Dear Authors,

The research written is interesting and in my opinion written quige well.

I have a few general comments:

Introduction: what were the times of curing of varnishes with light? The last sentence needs to be revised.

Methodology: what types of rollers for application were used? What was granulation by sanding? Impact resistance: hom much weight the weight, what was the diameter of ball and were the impact diameters measured?

Which sanding strips were used in abrasion test?

Line 271: the basecoat amont the stated also needs to assure adhesion between coating system and substrate.

Text in Figure 81(?) should be larger.

What mean the dots and crosses in Table 8?

Were the colour changes caused with cold liquids visible to naked eye? What is the minimum dE to detect them?

Thank you!

The English require minor correction.

Br

Author Response

Dear Reviewer,

first of all thank you very much for your kind words and useful comments about our paper. A linguistic correction has been carried out. English was checked by a native-speaker from the official office. Below our answer:

The research written is interesting and in my opinion written quige well.

I have a few general comments:

Answer:

Thank you very much for your kind words.

Introduction: what were the times of curing of varnishes with light?

Answer:

The curing times for coatings in the UV technologies are very short (2-4 seconds) and depend on several factors, e.g. line speed, number of layers of the coating, number of radiators, energy, etc.

The last sentence needs to be revised.

Answer:

The last sentence was deleted.

Methodology: what types of rollers for application were used?

Answer:

The used rollers were covered with 35 mm thick rubber-coated rollers with a hardness of 40 ShA.

What was granulation by sanding?

Answer:

We added information to the text: “The Heesemann LSM8+EA10 wide belt sander was used for the tests. Two abrasive belts with grain size P180-P220, with corundum coating were used. Then two layers of primer were applied. A further operation was sanding (abrasive belt with grain size P360-P400)”.

Impact resistance: hom much weight the weight, what was the diameter of ball and were the impact diameters measured? 

Answer:

Steel ball (C in figure 1) with a diameter of 14 mm and Rockwell hardness of 60 to 66 HRC, a rolling bearing ball for example (acc. to ISO 3290) were used. The largest diameter of the impact mark in each test area was measured with the use of a Brinell magnifier [43].

Which sanding strips were used in abrasion test?

Answer:

CS-10 abrasive wheels were used.

Line 271: the basecoat amont the stated also needs to assure adhesion between coating system and substrate.

Answer:

We agree with the reviewer. After next investigations in the next papers we will analyse interlayer adhesion between UV coating system and HDF substrate. After initial investigations delamination in the substrate (100% A acc. to the PN EN 4624 standard) was observed.

Text in Figure 81(?) should be larger.

Answer:

Thank you for your suggestion. The figure was prepared in the special program. When text will be larger the quality of the figure will be worse.

What mean the dots and crosses in Table 8?

Answer:

We used dots and crosses to describe the criteria (description can be found in the legend).

Were the colour changes caused with cold liquids visible to naked eye? What is the minimum dE to detect them?

Answer:

No visible changes were observed with the naked eye. In the next article (-s) we will analyse the colour changes after the accelerated aging test (with the use of colorimeter and naked eye).